# A New Digital Lake Bathymetry Model Using the Step-Wise Water Recession Method to Generate 3D Lake Bathymetric Maps Based on DEMs

**Siyu Zhu** [1,2], **Baojian Liu** [1], **Wei Wan** [1,*], **Hongjie Xie** [3], **Yu Fang** [2], **Xi Chen** [1], **Huan Li** [1], **Weizhen Fang** [1], **Guoqing Zhang** [4], **Mingwei Tao** [2] **and Yang Hong** [1,5,*]

1   Institute of Remote Sensing and GIS, Peking University, Beijing 100871, China; 13240779409@163.com or zhusy17@mails.tsinghua.edu.cn (S.Z.); liubaojian@pku.edu.cn (B.L.); chenxi928@pku.edu.cn (X.C.); huan.li@pku.edu.cn (H.L.); weizhenfang@pku.edu.cn (W.F.)
2   Hydraulic Engineering, Tsinghua University, Beijing 100083, China; fangyu516@gmail.com (Y.F.); tmw15@mails.tsinghua.edu.cn (M.T.)
3   Department of Geological Sciences, The University of Texas at San Antonio, San Antonio, TX 78249, USA; Hongjie.Xie@utsa.edu
4   Institute of Tibetan Plateau Research, Chinese Academic of Science, Beijing 100101, China; guoqing.zhang@itpcas.ac.cn
5   School of Civil Engineering and Environmental Science, University of Oklahoma, Norman, OK 73019, USA
*   Correspondence: w.wan@pku.edu.cn (W.W.); yanghong@ou.edu (Y.H.); Tel.: +86-10-6274-4775 (W.W. & Y.H.)

**Abstract:** The availability of lake bathymetry maps is imperative for estimating lake water volumes and their variability, which is a sensitive indicator of climate. It is difficult, if not impossible, to obtain bathymetric measurements from all of the thousands of lakes across the globe due to costly labor and/or harsh topographic regions. In this study, we develop a new digital lake bathymetry model (DLBM) using the step-wise water recession method (WRM) to generate 3-dimensional lake bathymetric maps based on the digital elevation model (DEM) alone, with two assumptions: (1) typically, the lake's bathymetry is formed and shaped by geological processes similar to those that shaped the surrounding landmasses, and (2) the agent rate of water (the thickness of the sedimentary deposit proportional to the lake water depth) is uniform. Lake Ontario and Lake Namco are used as examples to demonstrate the development, calibration, and refinement of the model. Compared to some other methods, the estimated 3D bathymetric maps using the proposed DLBM could overcome the discontinuity problem to adopt the complex topography of lake boundaries. This study provides a mathematically robust yet cost-effective approach for estimating lake volumes and their changes in regions lacking field measurements of bathymetry, for example, the remote Tibetan Plateau, which contains thousands of lakes.

**Keywords:** lake bathymetry; lake volume; DEM; slope

## 1. Introduction

Terrestrial lakes are an important component of global and regional water cycles since they contribute the most to, and are very sensitive indicators of, worldwide nonfreezing lake water storage [1]. Conventionally, the accurate calculation of lake water storage or volume relies on the correct mapping of lake bathymetry, which is the key parameter for lake morphology study [1–4]. However, in situ mapping of lake bathymetry is costly and labor intensive [5]. Additionally, due to water's high attenuation of electromagnetic waves, big lake bathymetry cannot be retrieved directly using satellite remote sensing [6]. Therefore, bathymetric maps for most of the lakes worldwide are

still lacking, except for a small number of lakes (e.g., Lake Ontario in the United States) that were measured mainly by sonar [5].

The most common and simple way of mapping lake bathymetry is to measure the point-by-point bathymetry over the lake surface by a bathymeter on a ship; a bathymetric map is then generated by interpolation. The mapping of lake bathymetry can be divided into two categories: the direct way (using sounding lines) and the indirect way (using sonars) [7,8]. The former was almost abandoned with the use of modern technology, and the latter is still widely applied using instruments such as the echo sounder, multibeam bathymeter, sub-bottom profiler, and seismic soundings [7,9]. The abovementioned methods can obtain acceptable and accurate lake bathymetric maps, which are inconvenient and expensive, especially for estimating the vast number of lakes regionally or globally. By utilizing a remote sensing approach, the capability of estimating lake volume in large areas can be improved [5,10]. Therefore, here we list some previously published methods for lake volume estimation:

1.  Airborne observation: This method uses airborne gravity data to determine the lake bottom shape with the assumed density of sediment [11]. The advantage of this method is that it can generate the sediment layer due to the density variance; the disadvantage is the high cost of airborne equipment for large-scale observations [11].
2.  Passive imaging method: This method is widely applicable to shallow turbid water bodies by the different reflectances of different wave bands [12]. There are two main ways to establish the relationship between the depth and reflectance (i.e., the equation fitting method) [13] and the machine learning method, such as ANN (artificial neural network) [14]. However, a necessary condition for this method is that the water must be shallow enough; it does not work for large and deep lakes [15].
3.  Empirical equation methods: These methods establish a formula for estimating the average depth with the lake area based on a large number of real average-depth data to calibrate the parameters [16]. Some other method even takes more information into account such as surrounding average slope [1]. However, they cannot generate bathymetric maps or derive the volume-depth and area-depth curves for single lakes.
4.  Similar volume–area (V–A) curve method: This method is based on the regional similarity of volume–area relationships. The method chooses several similar valleys as virtual reservoirs and then simulates the process of water filling and regression. It may be successful for researching small mountainous lakes, but for medium and large lakes, similar valleys are always lacking [17].
5.  Lake bathymetric map simulation method: This method uses the surrounding digital elevation model (DEM) data to estimate the lake bottom shape. For example, Messager et al. tested a GIS-based method based on power functions in which the depth is derived as ((the distance from shore)^$\alpha$ × tan(slope)) using different values of the exponent $\alpha$ [1]. We reproduced this method and found that the model result cannot maintain continuity, which is the most challenging barrier in the model proposed in this study. We finally overcame this problem by proposing a water recession method (WRM).
6.  The area–elevation combined methods: These methods use optical satellite imagery such as Landsat [18] or MODIS [19,20] to derive lake-surface area series; meanwhile, they use satellite altimetry such as ICESat/GLAS and ENVISAT [18,19] to generate surface elevation. The area and surface elevation series are then combined to calculate relative lake volume estimates. These methods are widely acceptable since they are based on simple and direct mathematical logics. However, the lack and disconnection of lake level data from satellite altimetry is a key issue that limits the applications of the abovementioned methods [18,21]. Besides, these methods can achieve accurate results of relative lake volume rather than absolute volume or bathymetry.
7.  Other interpolating methods and derivative spline methods: These methods basically use the control points in the calculation area and the control conditions of boundaries to generate the whole bathymetric map for water bodies, such as Kriging [22] or spline interpolation [23]. There are

many successful researches about rivers because of the regularity of the river bathymetry [24,25], and there are even some mature methods in ArcGIS tools [26], but the common point is that they need field survey results as input data. We admit the measured data is indispensable for high accurate research, and more uniform distribution of the input data will lead to more accurate results. However, in considering the big cost of large-scale in-situ data [5], it is still important to develop remote sensing methods without these data. This is also the starting point of our method.

This study develops a new digital lake bathymetry model (DLBM) to construct a lake bathymetric map using the DEM (digital elevation model) alone. The DLBM can generate an estimated 3D bathymetric map and benefits from the WRM, which overcomes the discontinuity problem to estimate the complex topography of the lake boundary. The model is written in MATLAB and is fully automated and available upon request.

## 2. Data and Model Development

This study's objective is to develop and validate a new cost-effective method that generates the lake volume/area trend along with the depth change to calculate the whole volume by using a purely DEM-based mathematical modeling approach as described below.

### 2.1. Input Data

The proposed model in this study is a mathematical model with DEM data as the input data. Here, the SRTM (Shuttle Radar Topography Mission) DEM data with a 90-m spatial resolution is chosen to execute the model and show the experimental results. However, other DEM data, such as airborne data, are also suitable as inputs. The SRTM DEM is originally produced by NASA and provides a major breakthrough in digital mapping of the world [27,28]. Although the 30-m version of SRTM is available, to make it consistent with the measured data from the National Oceanic and Atmospheric Administration (NOAA), we used SRTM-90 for experiments in this study.

### 2.2. Assumptions

Our DLBM is based on two main assumptions, as shown in Figure 1:

1.  The natural surface of the lake's bathymetry is typically formed and shaped by geophysical processes similar to those that shaped its surrounding landmass.
2.  The agent rate of water is uniform throughout the whole lake.

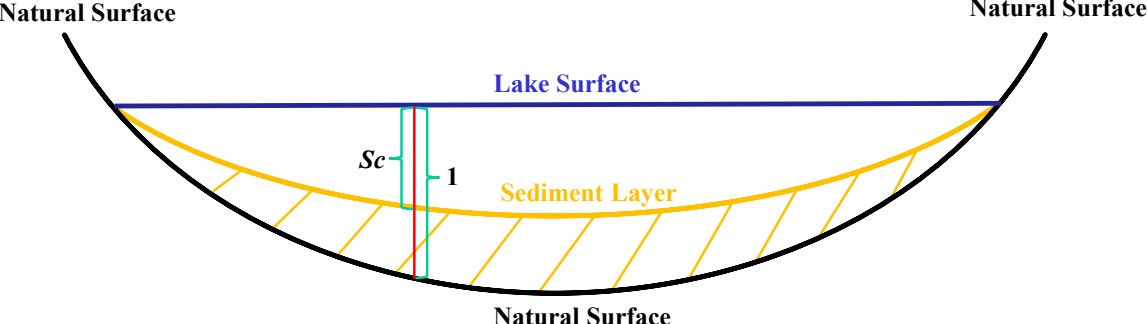

**Figure 1.** The deposition cross-section of the lake profile. The blue line represents the lake water surface, the black line represents the natural bottom of the lake, and the yellow area represents the sediment layer. The *Sc* is a scale value, equal to the water depth divided by the distance between the lake's water surface and the natural bottom. The *Sc* becomes 1 when there is no sediment.

Topography is the result of long-term processes (>1 Ma), and at this time scale, tectonics, geology, and climate are the main geomorphological determinants [29]. The geologic agent is defined as the

function that could cause geological effects, and it contains two important functions: erosion and sedimentation [30,31]. As shown in Figure 1, there is always a sediment layer covering the bottom of an arbitrary lake [32]. The uncertainty of the sediment layer becomes a common barrier for predicting the average lake depth [1]. We divided the lake bottom into two surfaces: a sediment layer surface and a natural surface. The real lake bathymetric map measured by sonar is closer to the sediment layer than the natural surface in the water [33]. Detailed explanations of the second assumption are shown in Appendix A.

Wind agents and water agents are the two main types of geologic agent [34–36]. The natural ground surface is mainly influenced by wind agents, and the sediment layer surface is mainly influenced by water agents [33,37,38]. By applying assumption (1), the bathymetric map generated by our model is theoretically the natural surface, which means we can use the surrounding slopes as the initial condition to simulate the lake's bathymetry; subsequently applying assumption (2), the thickness of the sediment layer is directly proportional to the water depth due to the uniform deposition rate. Based on these two steps, we propose the water agent rate, *Sc*, which equals the water depth divided by the distance between the lake surface and the natural surface. If the *Sc* is larger than 1, the erosion function of the agent is more prominent than the deposition function.

## 2.3. Model Procedure

The flow chart of the proposed DLBM is shown in Figure 2. The full name/definition of each abbreviation in the model procedure is shown in Table 1. The calculation is based on the water recession method (WRM). The WRM is an imitation of water recession, which means that the whole calculation process is similar to the process in which the lake gradually dries up as the water level decreases. The DLBM requires DEM data (with/without a real lake bathymetric map) as an input, and the estimated bathymetric map of the lake is the final output.

**Table 1.** The full name/meaning of the abbreviations in the model procedure.

| Abbreviation | Full name | Definition |
|:---:|:---:|:---|
| WRM | Water Recession Method | It is a process simulating the water recession. In this process, the water keeps decreasing and land appears gradually. |
| SSM | Surrounding Slope Module | It is a program module designed for calculating the surrounding slope of four directions |
| SSD | Surrounding Slope Data | It is the result of the SSM recorded by a m × n × 4 matrix. The value is slope, m and n represent position, and 4 represents direction. |
| LBM | Lake Binary Matrix | It is a binary matrix to record water and land by 1 and 0, and it has the same size with input from the digital elevation model (DEM). |
| H | Height | It is the current calculation of elevation, as well as the assumed water surface height in the WRM. |
| AGA | Already Generated Area | It is a binary matrix to record the calculated and uncalculated pixels, and it has the same format as that of the LBM. |
| DSP | Decreasing Step Parameter | It is the decreasing height of every single iteration and it controls the decreasing speed of the water surface in the WRM. |
| CCP | Current Calculation Point | The pixel which will be assigned to an estimated value in this present iteration. |
| BOD | Border-On Direction | In this direction, the CCP is adjacent to the water pixel rather than the land pixel. |
| MFM | Morphologic Function Module | It is a program module designed to calculate the estimated value of the CCP by mathematical function and input information. |
| *Sc* | Sediment coefficient | It is a correction factor serving for this model based on the two assumptions. |

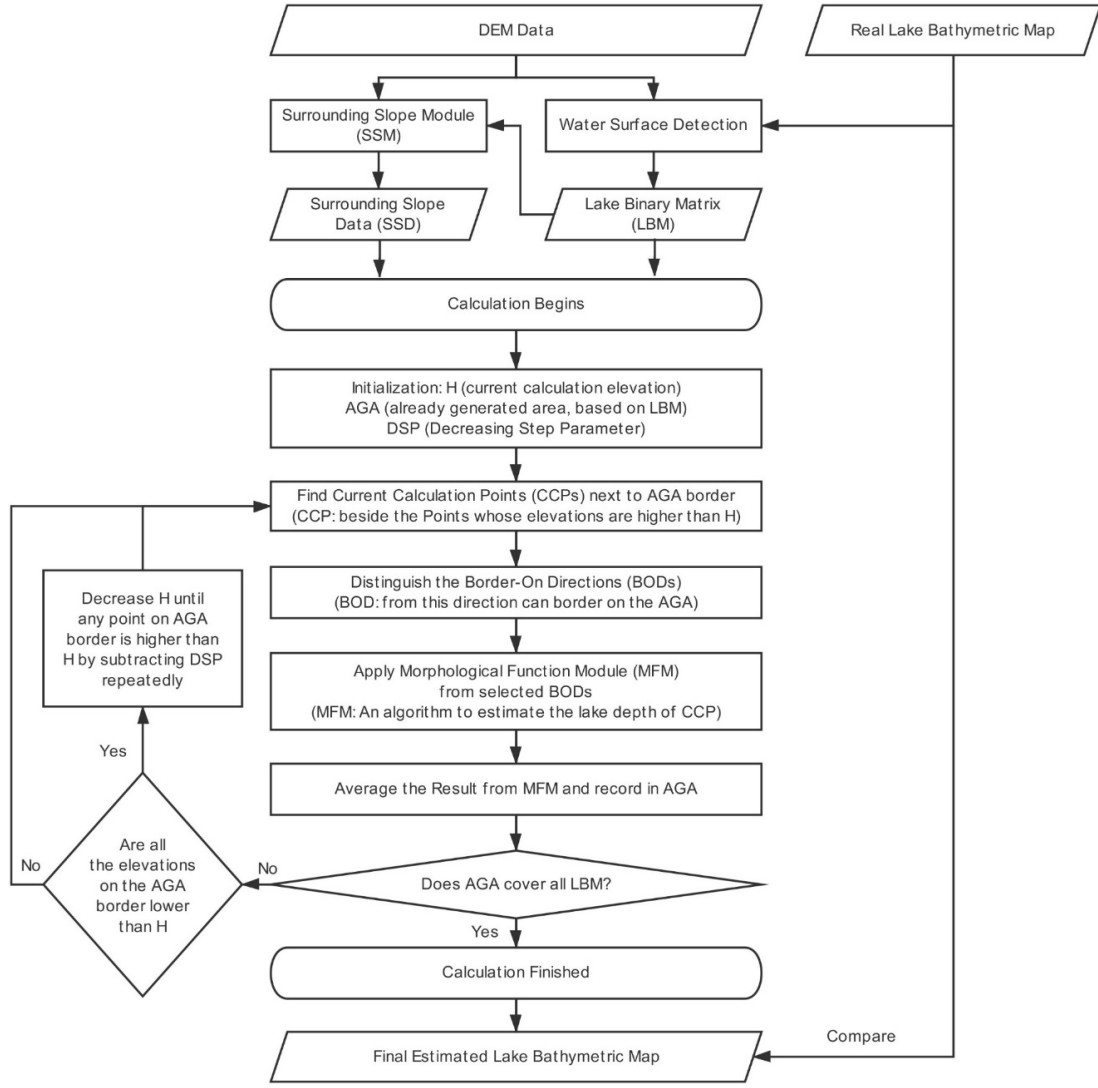

**Figure 2.** The flow chart of the proposed DLBM (digital lake bathymetry model).

Preprocessing is needed to generate two data sets: (1) the lake binary matrix (LBM) and (2) the surrounding slope data (SSD). First, we use DEM data to generate the LBM by water surface detection. The LBM has the same size and resolution binary matrix as the DEM data. The LBM records the lake location by setting the lake surface as 1 and the surrounding area as 0. If there is a real lake bathymetric map, we can use the lake boundary information in the DEM data to draw the outline and then binarize it. If there is no real lake bathymetric map, we may regard the pixel whose value is equal to the lake level as the lake surface (set as 1, others as 0) and then apply a filter function (a 3-pixel image corrosion and dilation) to delete those incorrect pixels (mainly including the points outside the lake boundary). Second, we generate the SSD by using the surrounding slope module (SSM, more details in Section 2.4).

After preprocessing, we need to initialize three important temporary variables: (1) height (H), set as the current calculation elevation, (2) the already generated area (AGA), set at the same size and resolution binary matrix as the LBM. The AGA records the already generated area as 1 (for uncalculated pixels) and 0 (for calculated pixels). The AGA corresponds to the area between the calculation bank (green line) and the original bank (black line) in Figure 3, and (3) decreasing step parameter (DSP), a parameter used to control the decreasing step of H.

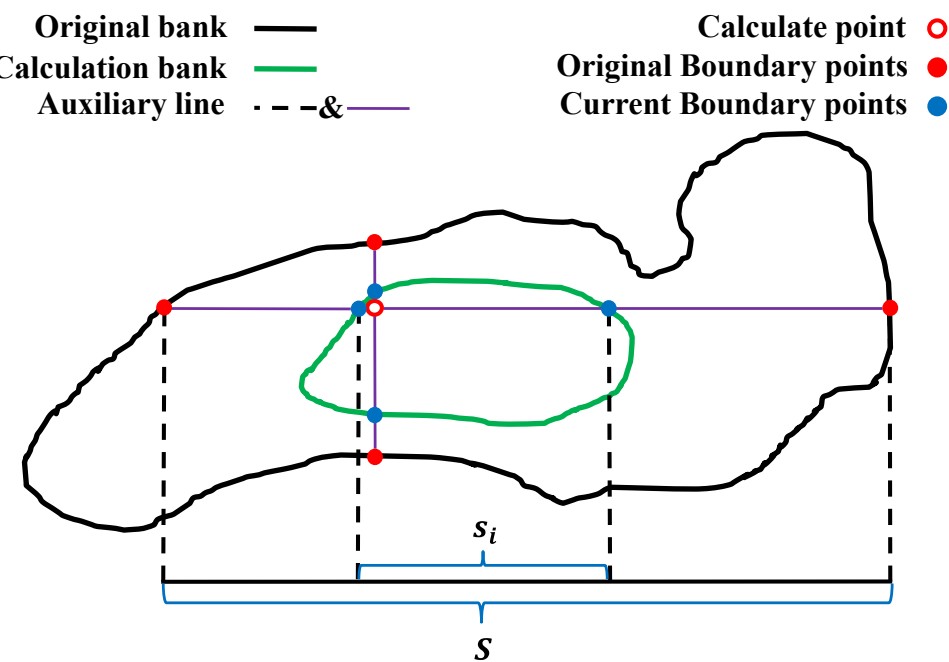

**Figure 3.** Model calculation procedure diagram. The black line represents the original bank of the lake, and the green line represents the calculated bank of the lake. The red circle is the current calculated pixel, and the red/blue solid dots represent the linked original/current boundary points. $S$ and $s_i$ represent the original and current calculation widths, respectively, which can be shown in the profile in Figure 5.

The circulation to calculate the estimated depth of the detected current calculation points (CCPs) is called an iteration. It includes the following steps: (1) finding CCPs, (2) determining border-on direction (BOD) of CCPs, (3) estimating the depth value of CCPs by the morphologic function module (MFM), and (4) updating the AGA and H. The iterations are executed until all pixels in the lake are assigned to the estimated depth values.

1.  Finding CCPs: The current calculation points (CCPs) are found through a circulation step by (1) drawing the AGA border adjacent to the uncalculated side, (2) finding the pixels whose values are higher than the H, and (3) defining the pixels that are in the uncalculated area but next to the pixels found in step 2 as the CCPs. For example, the hollow red point in Figure 3 is a CCP ready for calculation. Note that the CCP is the pixel that is calculated during each circulation interval.

2.  Determining BOD of CCPs: After the CCPs are found, we need to calculate each CCP based on the same AGA. For one CCP, before we apply the calculation formula, we need to distinguish the border-on directions (BODs), defined as the directions on which we can find a border-on AGA pixel. There are four potential directions (north, west, south, and east) for each CCP, and the BODs are defined as the directions on which we can find a border-on AGA pixel. As shown in Figure 3, the north and west directions are the BODs for this CCP. The reason for finding the BODs is based on the inverse distance weighted interpolation method [39], meaning that the influencing power decreases as the distance increases. In this model, we consider that the far coast has little influence on nearshore pixels, so the pixel elevation is only decided from the BODs. Obviously, the last pixel should have four BODs, while most cases have one to three BODs.

3.  Estimating the depth value of CCPs by the MFM: After the BODs are determined, we use the morphologic function module (MFM) to calculate the estimated pixel elevation from all BODs and then average the results (please see Section 2.5 for more details). This step iterates until the following situation is met.

4.  After obtaining the value of the CCPs, by now, all CCPs are assigned an estimated value in this iteration, the AGA and H will be updated to the next iteration. For the AGA, the position of

the CCPs in this iteration will turn to "calculated" from "uncalculated". As for H, after step 3, if all elevations of the AGA boundary drop below H, we then need to decrease H to make the calculation continue by a predefined decreasing step parameter (DSP); if not, it will do another iteration using the same H. In the former case, the new H equals the original H minus the DSP, and this process is continued until any CCP occurs. The DSP has a low sensitivity when it is small enough, but it also influences the efficiency of the program if it is too small. In this study, we set the DSP as 1 m in all tests with the same vertical resolution as the SRTM data. The H and DSP changing mechanism are the core of the WRM, and it makes the calculation progress relying on the water surface decrease rather than simply shrink the geometry boundary; this is better since the WRM imitates a natural process that makes the calculated surface almost horizontal rather than irregular.

### 2.4. Surrounding Slope Module (SSM)

The aim of the SSM is to generate the slope value of the surrounding lake boundary, including midlake islands. The SSM also has four directions that are in opposite directions to the corresponding BODs. For example, in Figure 3, the BODs of the CCP are north and east, and the slope directions of the northern point and eastern point are south and east, respectively. Here, we take the southern direction as the example shown in Figure 4. First, we set a slope radius parameter defined as the calculation radius of the DEM matrix. As shown in Figure 4, the slope radius is set as three for drawing convenience, and the middle pixel is the original boundary point (red points in Figure 3). To calculate the southern slope of the middle pixel, we apply Equations (1) and (2).

$$P(i) = (DEM(land(i)) - DEM(lake(i)))/cellsize \tag{1}$$

$$k1 = \sum_{i=1}^{n} P(i)/n \tag{2}$$

where $P(i)$ represents the temporary slope at location $i$; $DEM(land(i))$ and $DEM(lake(i))$ denote the adjacent elevation of the land and lake, respectively; *cellsize* is determined by the resolution of the DEM data, 90 m in this case; $k1$ is the final value of the southern slope, and $n$ is the total number of $P(i)$.

**Figure 4.** Slope calculation diagram, with land and lake pixels marked as black and blue, respectively. The red point is the slope calculation point, whose location is shown in a plan view in Figure 3. The $P(i)$ is the slope from the land to lake as indicated by the red arrow symbol, with the black arrow symbol on the bottom showing the direction of calculation. $k1$ is the final average slope value, which is calculated from Equation (2).

It is worth mentioning that the boundary may exceed the slope matrix (such as the left column in Figure 4) for which it is not considered in the slope calculation. In the tests of this study, we set the slope radius as 10 to reduce the possibility of this situation while enhancing the whole continuity of the bathymetric map. Meanwhile, if $k1$ was an unreasonable value (equal to or lower than 0), we revised it as the min-slope parameter. In our study, we set it as $1/cellsize$ (i.e., 1/90).

### 2.5. Morphologic Function Module (MFM)

The morphologic function module (MFM) is the core of the proposed model since it directly controls the morphology of the final result. The purpose of the MFM is to calculate an estimated value of one CCP based on the AGA and SSD.

The calculation diagram of the MFM is shown in Figure 5. Here, the AGA is the side area between the blue points and red points; the yellow point is an assumed lake bottom point of this profile. It is obvious that the slope of the bottom point must be 0, although we do not know the exact location; others can be referred to in the caption of Figure 5. It is necessary to explain the tiny difference between the elevations of the current boundary points due to the WRM and DSP, since the pixel values are recorded by double float rather than integer, making the decimal part not exactly equal. However, this difference is too small and can be ignored (i.e., $k0_i = 0$).

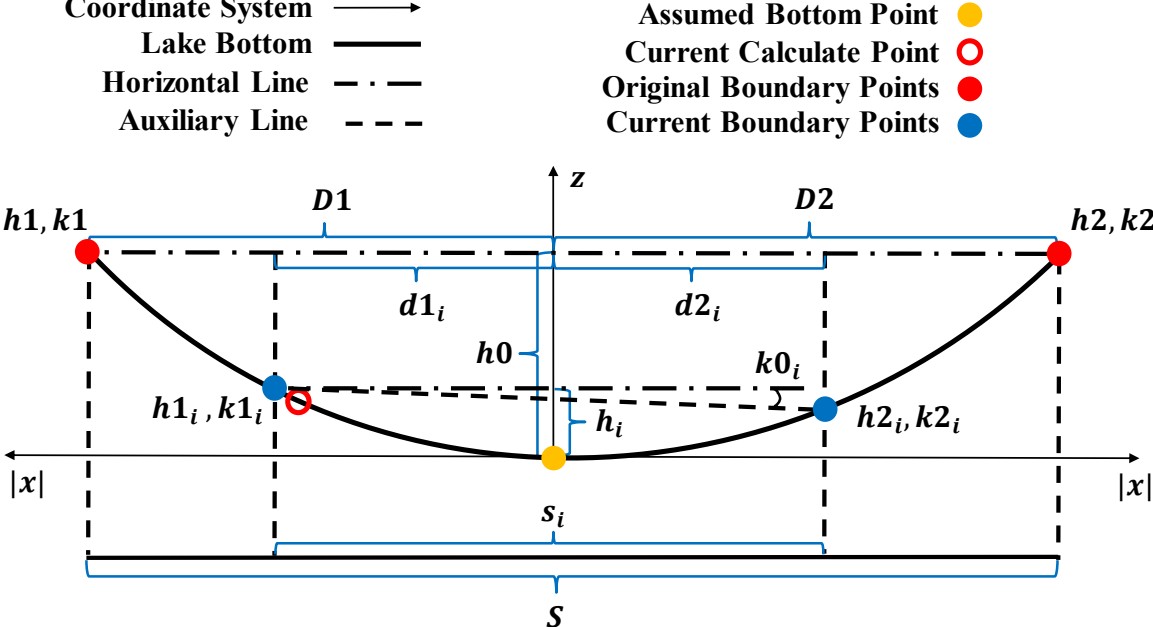

**Figure 5.** The calculation diagram of the MFM (morphologic function module). The black line represents the estimated lake bottom. The red circle is the pixel which is calculated now, and the red/blue solid points represent the linked original/current boundary points (same as in Figure 3). The yellow point represents the assumed bottom of the lake with zero slope. $h1$ and $h2$ are the elevations of original boundary points (actual lake surface level with the same values); $h1_i$ and $h2_i$ are the elevations at current boundary points, meanwhile they are both assumed to be equal to $h_i$ due to small $k0_i$ values. $k1$ and $k2$ are slope data with the corresponding direction from the SSD (Figure 4); and similarly, $k1_i$ and $k2_i$ are the slopes at current boundary points through calculation. Other parameters: $S$ and $s_i$ represent the original/current calculation width, respectively; $h0$ and $h_i$ represent the original/current calculation depths, respectively; $D1$ and $D2$ represent the horizontal distance between the assumed bottom and original boundary points; similarly, $d1_i$ and $d2_i$ represent the horizontal distance between the assumed bottom and current boundary points. Furthermore, the $k0_i$ is a small decimal deviation because the data is recorded by double float format.

To calculate the elevation of the CCP (hollow red point), we need to know the elevation and slope of the adjacent point (left blue point) and then apply Equation (3).

$$Z(i) = h1_i - k1_i * cellsize \tag{3}$$

where $Z(i)$ denotes the referenced elevation of the calculated point, $k1_i$ denotes the slope value (positive) of the calculated point, and *cellsize* is just as previously mentioned in Equation (1).

We assume that the slope decreases from $k1$ and $k2$ (original boundary) to zero (assumed bottom point) following a simple function along the lake bottom line of the profile. Therefore, we assume that $k$ is a function of $d_i/D$. We set a coordinate system on the assumed bottom point, where $x$ denotes the distance from the assumed bottom point and $z$ denotes the elevation. Based on this, there are two main formulas to calculate $k1_i$, as shown in Equations (4) and (5).

$$k(x) = k0 \times \left(\frac{x}{D}\right)^n \tag{4}$$

$$k(x) = k0 \times \sin\left(\frac{x}{D} \times \frac{\pi}{2}\right) \tag{5}$$

where $x$ is the horizontal ordinate; $D$ is the horizontal distance between the original boundary point and the assumed bottom point; $k0$ is the slope on the original boundary point from the SSD; and $n$ is a variable parameter that is 1 or 2, ordinarily.

$$k1_i = k1 \times \left(\frac{d1_i}{D1}\right)^n = k1 \times \left(\frac{s_i}{S}\right)^n \tag{6}$$

$$k1_i = k1 \times \sin\left(\frac{d1_i}{D1} \times \frac{\pi}{2}\right) = k1 \times \sin\left(\frac{s_i}{S} \times \frac{\pi}{2}\right) \tag{7}$$

Due to the uncertainty of the assumed bottom point, $d_i/D$ is unknown, so we replace it with $s_i/S$ in morphology Equations (6) and (7). This replacement is strictly established in a two-dimensional situation (profile situation), which is further proven in Appendix B.

## 3. Model Validation

To test and validate the proposed model, we used two lakes as examples that represent two different situations. One is Lake Ontario, with the measured bathymetric data, and the other is Lake Namco, with a recorded water level, area, and volume series.

### 3.1. Analysis of Lake Ontario

Lake Ontario (77.9° W, 43.4° N) is the 14th largest lake in the world and one of the five Great Lakes of North America [40], with a water level of ~74 m, an area of ~18,960 km$^2$, and a volume of ~1640 km$^3$ (which equals to an average depth of 86 m) [41]. We downloaded the measured bathymetric lake data of Ontario from the National Oceanic and Atmospheric Administration (https://ngdc.noaa.gov/), as shown in Figure 6a,b. Figure 6 shows the lake bathymetric map and surrounding elevations at the same resolution as the SRTM-90 data. We extracted the lake boundary by these measured data and then applied it to the SRTM data for model processing. Note that the lake elevation in these data is zero, but in the SRTM data, it is 73 m. The difference may be due to the different reference ellipsoids used in these two data sets. The results in Figure 6c,d are derived from the scaled model based on the SRTM data (hereafter called model 1). For comparison, we conducted another control experiment using the measured bathymetric map (hereafter called model 2), which means that we used it as an alternative to the SRTM data. The results of model 2 are shown in Figure 6e,f.

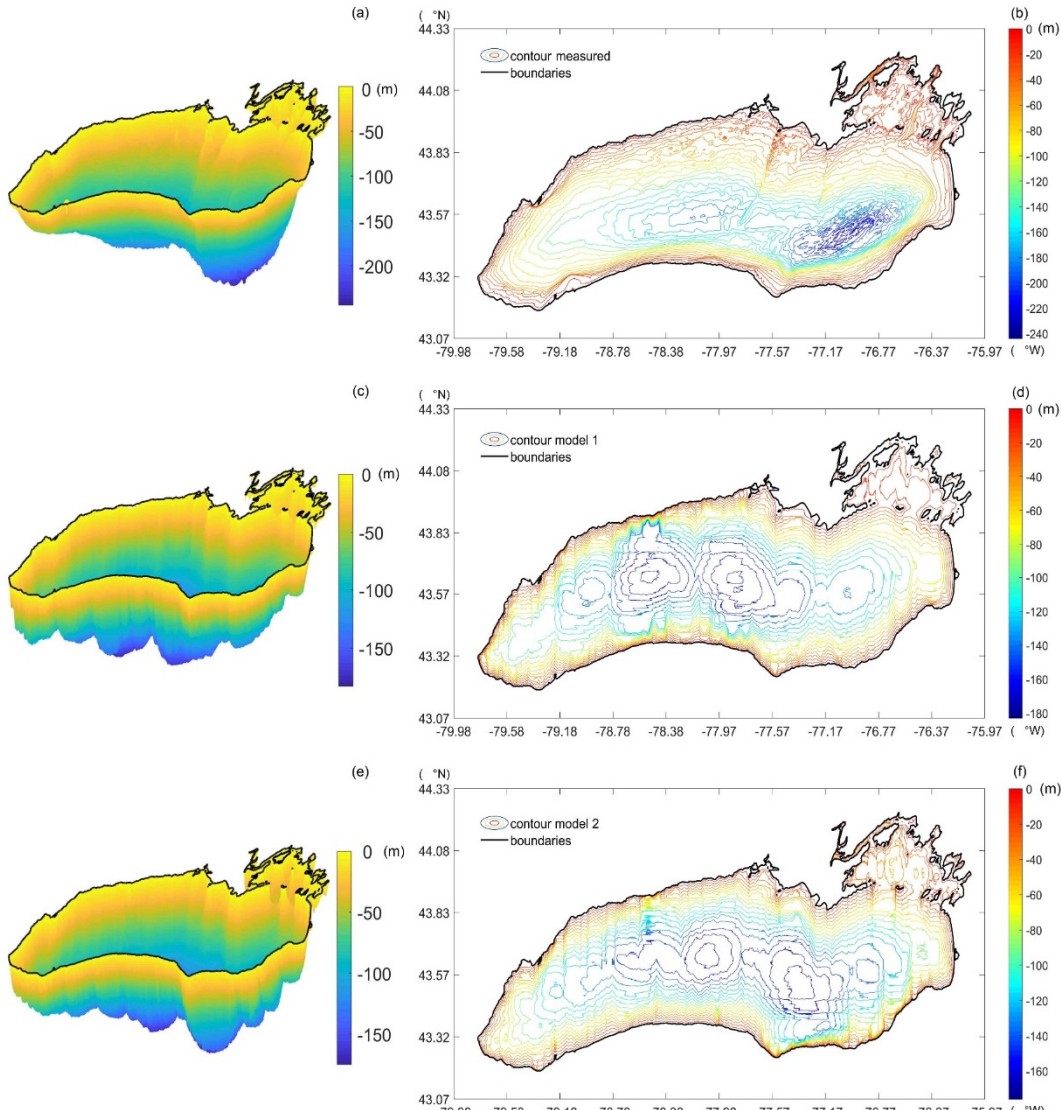

**Figure 6.** The modeling results of Lake Ontario. (**a**) 3D map of the measured bathymetric data; (**b**) contour map of the measured bathymetric data; (**c**) 3D map of the result using scaled model 1; (**d**) contour map of the result using scaled model 1; (**e**) 3D map of the result using scaled model 2; (**f**) contour map of the result using scaled model 2.

As mentioned before, the model result should be revised by the parameter *Sc* to remove the influence of deposition. Since the average depth of the measured bathymetric map is 88.58 m and the average depth from the original model 1/model 2 is 146.00 m/156.65 m, we derive the *Sc* as 0.61/0.57. After using the *Sc* to scale the original model result, we obtain the scaled model 1/model 2 maps, as shown in Figure 6c,d/e,f, which have the same average depth as the measured bathymetric map. What needs to be explained is that the graphic shapes of the original model result and the scaled model result are the same, and we just plot the latter for comparison.

From Figure 6c,d, we can calculate the bottom points of the lake, which occur at 77.0° W, 43.4° N (as the extreme bottom) and 78.2° W, 43.5° N (as the sub-extreme bottom). Model 1 shows three main bottom points with locations similar to those of the measured data, with the western/eastern extreme bottoms deeper/shallower than expected. This difference could be caused by the uncertainty of the SRTM data and the narrow range of the slope calculation length. Model 1 successfully captures some ridges, especially the one along the 77.6° W longitude line around 43.8° N; the ridge appears to be the rising feature around the islands that mainly exists in the northeast area. Model 2 successfully captures

this feature as well, and the two main extreme bottom areas are seen in Figure 6e and f at similar locations. The lake bottom derived from the two models looks gentle rather than rough and complex. The sub-extreme bottom is relatively overestimated, but the extreme bottom is underestimated after scaling. Meanwhile the extreme bottom of model 2 is more eastward and closer to measured data than model 1. This phenomenon explains that more accurate land data (e.g., the bathymetric map in model 2) will lead to more accurate results, since the surrounding slope directly determines the final shape of the lake bathymetry.

Figure 7 shows the area/volume-depth curve using Equation (5) as the shape-fitting equation. Due to scaling, the volume curves of the original and scaled models are similar, but the area curves are not. The scaling cannot achieve a consistent average depth and maximum depth at the same time. Therefore, for either model 1 or model 2, the maximum depth is overestimated in the original result but underestimated after scaling. Similarly, the volume curves of the scaled model fit well, but the area curves differ more. This means that the area is underestimated for deep water but overestimated for shallow water due to the integration-equal principle (equal to volume). The most satisfying thing is that the volume-depth curves fit well and the two lines of model 1 and model 2 are almost overlapping. This phenomenon demonstrates that area fitting is more difficult due to its high sensitivity, but volume fitting is more acceptable for a low sensitivity to the surrounding DEM data.

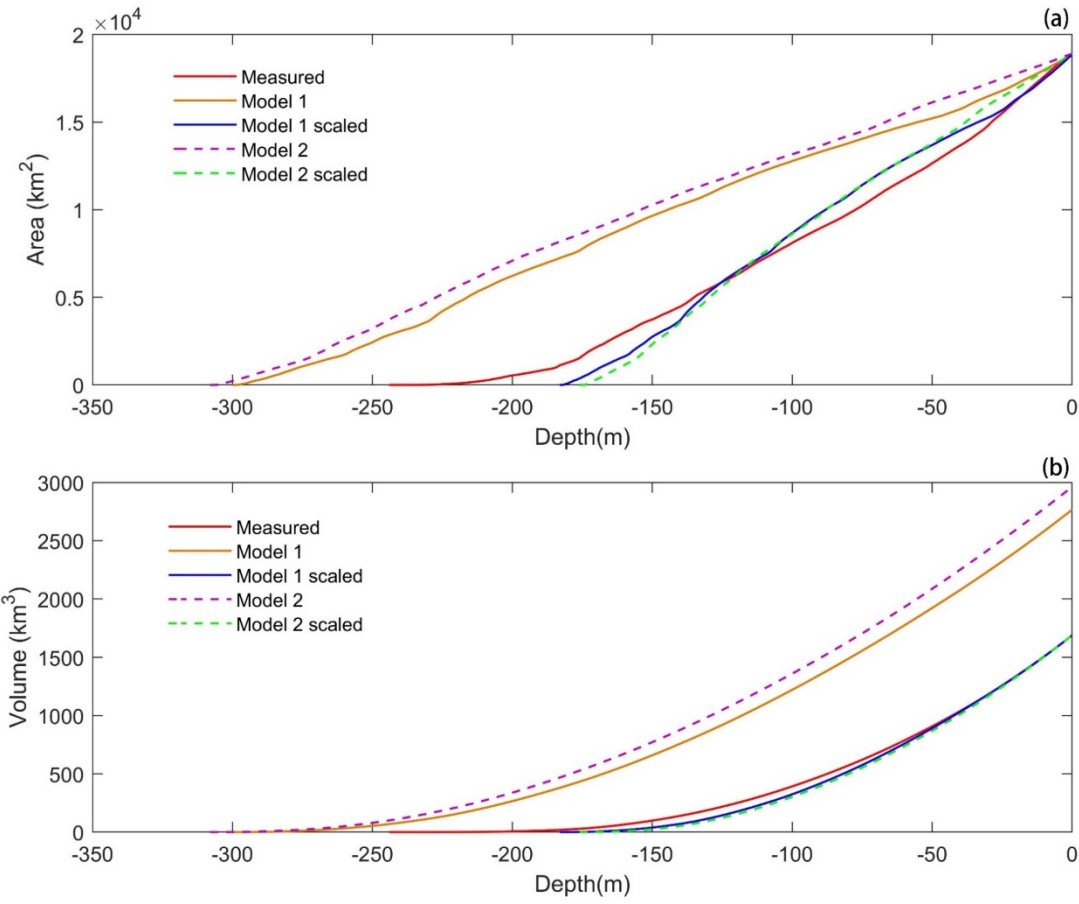

**Figure 7.** The (**a**) area-depth curve and (**b**) volume-depth curve of Lake Ontario in model 1 and model 2.

For statistical analysis, the mean absolute error (MAE), root mean square error (RMSE) for lake bathymetry, and R-squared for water component curves are calculated, and the formulas are listed below, respectively.

$$\text{MAE} = \frac{\sum_{i=1}^{n}\left|h_{model,i} - h_{measured,i}\right|}{n} \tag{8}$$

$$\text{RMSE} = \sqrt{\frac{\sum_{i=1}^{n}\left(h_{model,i} - h_{measured,i}\right)^2}{n}} \tag{9}$$

where $h_{model,i}$ represents the lake depth of $i$th pixel derived by model 1/model 2; similarly, $h_{measured,i}$ represents the lake depth of $i$th pixel in measured NOAA data; $n$ represents the total water pixel number.

$$\text{R-squared} = 1 - \frac{\sum_{i=1}^{n}\left(y_{model,i} - y_{measured,i}\right)^2}{\sum_{i=1}^{n}\left(y_{measured,i} - y_{measured,mean}\right)^2} \tag{10}$$

where $y_{model,i}$ represents the $i$th area/volume (A/V) value generated by model 1/2; $y_{measured,i}$ represents the $i$th A/V value generated by measured data; $y_{measured,mean}$ represents the mean value of $y_{measured,i}$ data series; $n$ represents the total number of data series.

The MAE and RMSE here are used to describe the uncertainty of lake bathymetry. It means that they are calculated from Figure 6d,f minus Figure 6b. R-squared is used to describe the goodness-of-fit between data series derived from the model and measured data, and in this study, we use R-squared for water component curves including A-h curve and V-h curve between the scaled model and measured data (shown in Figure 7). Note that these curves are generated by a lake depth series (h) with a 1-meter interval from zero. The final results of Lake Ontario are listed in Table 2.

**Table 2.** Statistical analysis result of Lake Ontario.

| Model | MAE | RMSE | R-squared of A [1] | R-squared of V [1] |
|:-----:|:---:|:----:|:------------------:|:------------------:|
| 1 | 24.232 m | 33.957 m | 0.982 | 0.993 |
| 2 | 22.076 m | 29.312 m | 0.973 | 0.988 |

[1] R-squared of area/volume (A/V) represents the R-squared calculated by area/volume series data. MAE, mean absolute error; RMSE, root mean square error.

As shown in Table 2, the MAEs are around 23 m and the RMSEs are ~31 m, and model 2 has a better performance than model 1. Compared to the average depth of 86 m, the deviation is 25–40%, which illustrates that the model results cannot replace the precise in-situ measurements (3% or even lower) [42]. Nevertheless, this accuracy is still acceptable due to the fact that there are no in-situ control points in the lake to calibrate the modeled bathymetry. As for water component curves, the R-squared values are overall ~0.98, which is a satisfied result since they perform a highly interrelated relation. This means we can use this model to predict the area and volume change in most cases. It is worth to explain that, as shown in Figure 7, the fitting of volume is better than that of area, which is consistent with the R-squared values. However, for both area and volume, model 2 has a lower MAE and RMSE but smaller R-squared than model 1. The reason is that model 1 has a deeper max depth which leads to a better R-squared. Therefore, it is rational that more accurate surrounding DEM could lead to more accurate results due to the negligible difference of R-squared.

### 3.2. Analysis of Lake Namco

Lake Namco (90.60° E, 30.73° N), at present, is the second largest lake in the Tibetan Plateau and the third largest salt lake in China (the area exceeded that of Lake Seling Co in 2014) [43,44], with a water level of ~4724 m, an area of ~1964 km$^2$, and a volume of ~87.1 km$^3$, which equals an average depth of 44.4 m [1]. There is no accurate lake bathymetric map for Namco. For comparison, we downloaded the water level/area/volume data series from the Hydroweb website (http://hydroweb.theia-land.fr/) provided by a French national interagency organization. Note that there is no area/volume data for Lake Ontario, thus we cannot perform the same analysis on Lake Ontario.

For the example of Namco, we still use Equation (5) as the morphology equation. As mentioned before, the average depth of Namco from the reference is 44.4 m, and the original average depth derived from our model is 67.1 m, which means the *Sc* should be set as 0.66. The final scaled result

is shown in Figure 8. An extreme bottom occurs at 90.58° E and 30.70° N. Note that the islands at 90.38° E, 30.83° N and 90.77° E, 30.83° N are reproduced well.

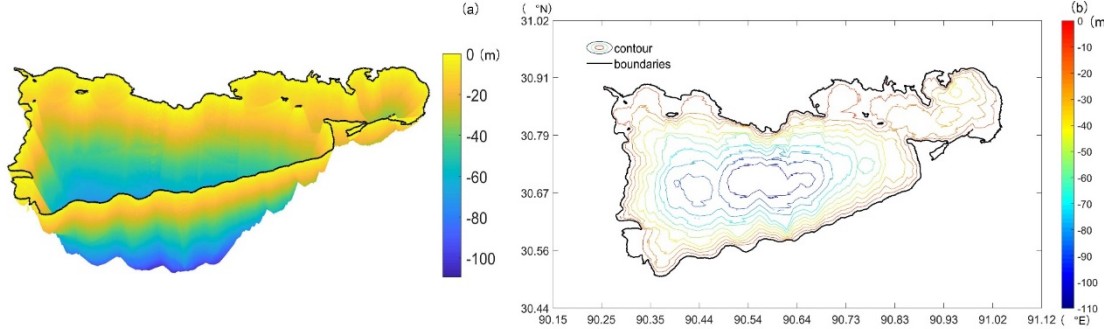

**Figure 8.** The modeling results of Lake Namco. (**a**) 3D map of the result using scaled model 1; (**b**) contour map of the result using scaled model 1.

Since there are no measured bathymetry data, we compared the modeling result with the observed data series from Hydroweb. The lake level data (Figure 9a) are the published data of Hydroweb from 2000 to 2017 (zero represents 4724 m), and we generated the lake area (Figure 9b) and relative lake volume (Figure 9c) using the scaled model, which applies the surrounding SRTM data and the Hydroweb lake level series. Note that the area series are absolute values, but the volume series are relative values, since the Hydroweb data do not give the absolute volume values. We chose a certain volume as the base value when its water level equals the volume-based water level (4722 m), which makes the relative volume of the Hydroweb data zero.

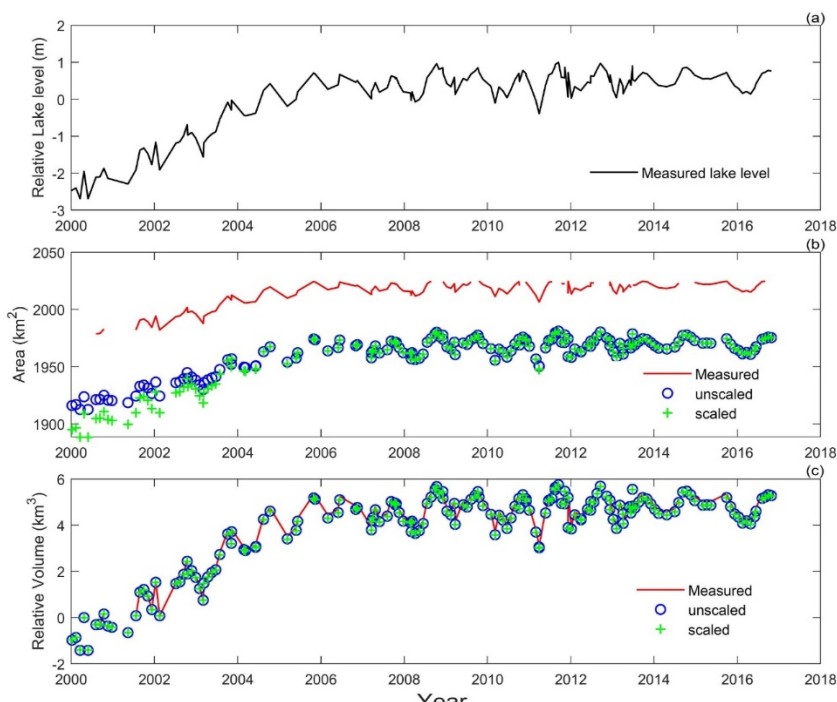

**Figure 9.** The (**a**) lake level from Hydroweb, (**b**) area, and (**c**) relative volume series result of the scaled model and reference data (from Hydroweb) from 2000 to 2017.

The area series result is unsatisfactory, with a difference of approximately 80 km$^2$. The area difference is a common issue when deriving lake areas using different data sources (e.g., in this study, SRTM, and optical remote sensing data). The area we derived is similar to the value published in the HydroLAKES dataset [1], since the SRTM is the data source for both methods. The area recorded in the

Hydroweb data is close to the value derived from Landsat data [45,46]. At present, it is recognized that lake areas derived from optical remote sensing imagery are more accurate due to the uncertainty of the SRTM; however, optical imagery may face distortion problems, especially on the edge of the image, which means that the derived lake area is inaccurate unless the image is georeferenced correctly [47,48].

Figure 10 shows the fitting results of the area and relative volume series. We found that they are almost distributed in a line, thus we applied a linear regression [49]. For unscaled results, the regression generates the fitting lines $y = 1.197x + 450.080$ for area and $y = 0.972x - 0.003$ for volume; as for the scaled model, the regression generates the fitting lines $y = 1.529x + 1121.222$ for area and $y = 0.970x - 0.008$ for volume. The two fitting lines for volume both have high regression coefficients close to 1, since the measured data and modeled data are derived using the same data source of lake elevations. Therefore, the regression coefficients are invalid, but the closeness to the equal line $y = x$ represents the goodness of fitting. Figure 10 also shows that, although with inaccurate area estimates, the relative volume estimates are good enough to accomplish a deviation of approx. 3%. This conclusion is also illustrated in the RMSEs as shown in Table 3. The RMSE of area exceeds 50 km$^2$, which means the areas from DEM and optical remote sensing have great discrepancies. However, the RMSE of volume for both cases are low (~0.13 km$^3$), which shows the potentiality of the proposed model used for volume estimates. As for the abnormal phenomena that RMSE of a scaled result is worse than that of an unscaled, the reason is that the *Sc* scaling is designed for lake bathymetry over large-scale, rather than for small, data series. More work is needed to obtain a satisfactory result with regard to the area series. Moreover, compared to lake volume, the lake area is much easier to derive from optical remote sensing data; thus, the importance of the area estimate is much lower than that of the volume estimate.

**Table 3.** RMSEs for area and volume of Lake Namco.

| Model | Area | Volume |
|---|---|---|
| Unscaled | 53.669 km$^2$ | 0.123 km$^3$ |
| Scaled | 55.896 km$^2$ | 0.135 km$^3$ |

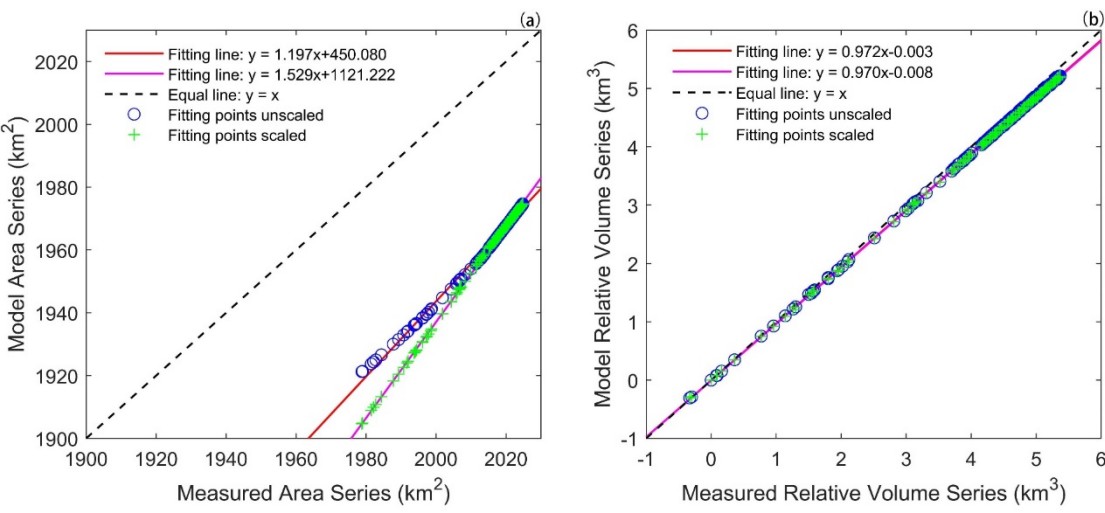

**Figure 10.** The fitting result of the (**a**) area and (**b**) volume series between the Hydroweb data and the proposed model.

## 4. Discussion and Conclusions

In this paper, we develop a new digital lake bathymetry model using surrounding slopes derived from DEM data. By proposing the water recession method, the generated 3D bathymetric map can overcome the discontinuity problem that existed in previous studies to adopt the complex topography of lake boundaries. We take Lake Ontario and Lake Namco as two examples to validate the model.

Note that the model has limitations in predicting the true lake bathymetry when it only uses the surrounding topography, given the uncertainty of the lake bottoms. Nevertheless, the proposed DLBM in this study is a feasible solution for applications such as estimating the lake volume.

Compared to traditional sonar methods or airborne gravity data, the proposed DLBM is more convenient and cheaper for estimating lake volume. Compared to the passive imaging method, the DLBM overcomes the conditional limitations of shallow turbid water bodies. Compared to the empirical equation method, the DLBM can generate a 3D bathymetric map rather than just one estimated value, and users can further derive the volume variability using a water level series, as shown by the example of Lake Namco in this study. Compared to other 3D modeling methods, the DLBM conquers the discontinuity problem and can adopt the complex topography of lake boundaries, including islands, with high robustness. Moreover, the DLBM can be applied to other, finer DEM data beyond the current SRTM data and their resolution.

One potential uncertainty of the DLBM is that it is sensitive to the surrounding slopes, which rely on the accuracy of DEM data. Another potential improvement of the DLBM is to add more geologic parameters to calibrate this model; for example, we could find a rule to predetermine the parameter *Sc* rather than just using a mathematically calculated value. We believe that there is a pattern between the *Sc* and the geologic elements. The geologic elements may include the cause of formation, the salinity of water, or the soil type of the region. For instance, the Great Lakes are the result of glacial erosion [50], and Lake Namco is a famously salty lake with high levels of Mg and Ca [51].

To summarize, the DLBM approach proposed in this study is a mathematically adventurous attempt since it estimates unknown quantities using very limited known quantities. Though it cannot be a total replacement of in situ measurements, this study provides a cost-effective method for a good faith estimation of lake bathymetric maps over regions lacking field measurements of bathymetry, for example, the remote Tibetan Plateau, which contains thousands of lakes.

**Author Contributions:** We appreciate the work made by every author, and without their help the work would hardly have been done. S.Z. is the major contributor of this work including the method design, program and manuscript, meanwhile B.L. helped to improve the program efficiency which is important for batch processing. Y.H., H.X., and W.W. were the instructors of the experiment to determine the method, meanwhile they also wrote the background/conclusion sections and helped to revise the whole manuscript. The other authors also offered the necessary help with data, parts of the program, the discussion, or sections of the manuscript. For more details, Y.F. and H.L. offered and processed the data from Hydroweb. W.F. and X.C. helped program the water body identifying section on DEM. M.T. and G.Z. helped download and process DEM data to accomplish the automatic montage function.

**Funding:** This study is jointly supported by the Key R&D Program of the Ministry of Science and Technology, China (Grant No. 2018YFC1506504) and the National Natural Science Foundation of China (NSFC) (Grant No. 91437214 and No. 41501360).

**Conflicts of Interest:** The authors declare no conflicts of interest.

## Appendix A

Figure A1 shows the common process of how a lake is formed. In most cases, a lake is formed after water pours into a natural basin (Figure A1a). We assume that the turbidity of water is the same everywhere within the lake (Figure A1b). Under the function of gravity, the silt and other impurities begin to deposit, and the sediment thickness is proportional to the original depth due to the average distribution of silt (Figure A1c,d). This is the foundation of the second assumption. However, practically, the process shown in Figure A1 is not the end. As we know, whether the material will slide down on the slope or not depends on gravity and friction (Figure A1e). The critical slope is called the angle of repose [52]. Therefore, the final shape keeps changing because the part whose slope is bigger than the angle of repose will drop down to the lower position (Figure A1f). The angle of repose under water is basically around 0.3, and in our model and most cases, the terrain slope is much lower than [53]. Therefore, we omit that and use (Figure A1d) as the final assumption.

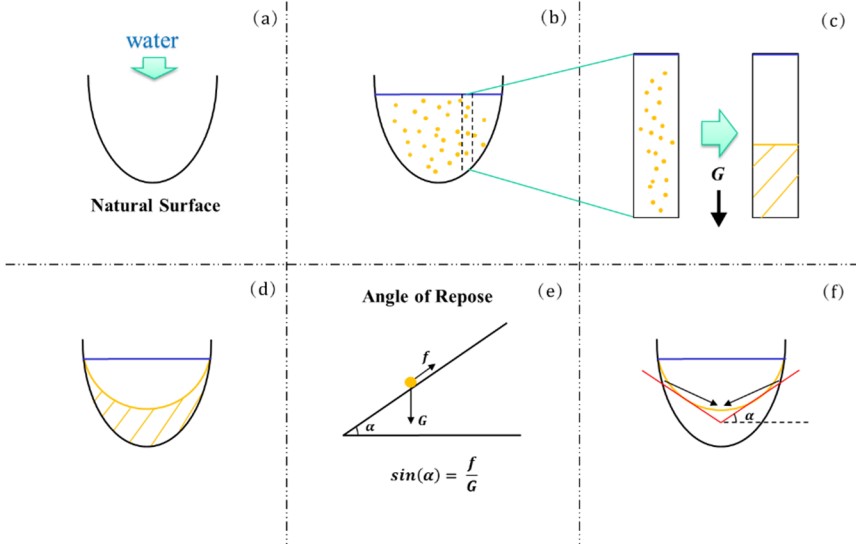

**Figure A1.** The common process of how a lake is formed. The black line represents the natural surface; the yellow line/points represent the sediment surface/material; and the blue line represents the water surface.

## Appendix B

Formula derivation for Equation (6):

First, we integrate the $k(x)$ function to obtain the profile curve function $z(x)$:

$$z_1(x) = \frac{k1 \times D1}{n+1} \times \left(\frac{x}{D1}\right)^{n+1} \tag{A1}$$

where $z_1(D1) = h0$, then

$$h0 = \frac{k1 \times D1}{n+1} \ \& \ z_1(x) = h0 \times \left(\frac{x}{D1}\right)^{n+1} \tag{A2}$$

by $z_2(D1) = h0$, and by the same logic

$$h0 = \frac{k2 \times D2}{n+1} \ \& \ z_2(x) = h0 \times \left(\frac{x}{D2}\right)^{n+1} \tag{A3}$$

with the condition $z_1(d1_i) = h_i \ \& \ z_2(d2_i) = h_i$

$$z_1(d1_i) = h0 \times \left(\frac{d1_i}{D1}\right)^{n+1} = h0 \times \left(\frac{d2_i}{D2}\right)^{n+1} = z_2(d2_i) \tag{A4}$$

Finally, we obtain

$$\frac{d1_i}{D1} = \frac{d2_i}{D2} \tag{A5}$$

according to the proportions-sum theorem

$$\frac{d1_i}{D1} = \frac{d2_i}{D2} = \frac{d1_i + d2_i}{D1 + D2} = \frac{s_i}{S} \tag{A6}$$

Formula derivation for Equation (7):

Using a similar step, we integrate the $k(x)$ function to obtain the profile curve function $z(x)$:

$$z_1(x) = C - \frac{2 \times k1 \times D1}{\pi} \times \cos\left(\frac{x}{D1} \times \frac{\pi}{2}\right) \tag{A7}$$

where $C$ is an integral constant.

With $z_1(D1) = h0$ & $z_1(0) = 0$, then

$$C = h0 = \frac{2 \times k1 \times D1}{\pi} \ \& \ z_1(x) = h0 - h0 \times \cos\left(\frac{x}{D1} \times \frac{\pi}{2}\right) \tag{A8}$$

by $z_2(D1) = h0$ & $z_2(0) = 0$, and by the same logic

$$C = h0 = \frac{2 \times k2 \times D2}{\pi} \ \& \ z_2(x) = h0 - h0 \times \cos\left(\frac{x}{D2} \times \frac{\pi}{2}\right) \tag{A9}$$

with the condition $z_1(d1_i) = h_i$ & $z_2(d2_i) = h_i$

$$z_1(d1_i) = h0 - h0 \times \cos\left(\frac{d1_i}{D1} \times \frac{\pi}{2}\right) = h0 - h0 \times \cos\left(\frac{d2_i}{D2} \times \frac{\pi}{2}\right) = z_2(d2_i) \tag{A10}$$

Finally, we obtain

$$\frac{d1_i}{D1} = \frac{d2_i}{D2} \tag{A11}$$

according to the proportions-sum theorem

$$\frac{d1_i}{D1} = \frac{d2_i}{D2} = \frac{d1_i + d2_i}{D1 + D2} = \frac{s_i}{S} \tag{A12}$$

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
