# Peer review of "A New Digital Lake Bathymetry Model Using the Step-Wise Water Recession Method to Generate 3D Lake Bathymetric Maps Based on DEMs"

_water, doi:10.3390/w11061151_

Round 1

Reviewer 1 Report

The manuscript is an interesting study presenting the new method of generating lake bathymetry based on DEMs.

The proposed methodology is a new approach to solving the problem of obtaining bathymetric map without expensive and difficult terrain measurements. The authors proposed new lake bathymetric model using the step-wise water recession method based only on digital elevation model. This is a very interesting approach because we do not need complicated input data to create a lake bathymetric maps. In this approach, two preliminary assumptions were made: the lake and its surroundings were formed by the same geological processes; the agent rate of water was uniform.

The model was examined in two cases - a lake for which a conventional bathymetric measurement was carried out (Lake Ontario) and a lake for which there is no accurate bathymetric map (Lake Namco). The proposed approach may be helpful in estimating bathymetry of lakes for which there is no accurate bathymetric measurements or their executing in the field is complicated.

The introduction presents the problem of acquiring accurate lake bathymetric maps. Such measurements are expensive, they can not always be performed due to difficult terrain conditions. Not all methods work well for all lakes (shallow - deep, transparent water - turbid water, etc.). The authors also very well presented earlier approaches to lake volume estimation. The introduction presents a sufficient literature review of analyzed issue.

An important and positive elements in the proposed method are:

-          the model is fully automated and available upon request,

-          the user needs only DEM data to generate 3D lake bathymetric maps.

The article has discussed in detail the structure of the model and its operation, however the descriptions of some components are complicated and relatively difficult to understand.

For the readers who follows the model's operation scheme, it may be difficult to understand it, when some of components (SSM, MFM) are described in further sections. The reviewer understands, however, that a clear presentation of the whole procedure is complicated when it contains many elements that need to be clarified. Maybe you should consider a slightly different structure of this part of the manuscript?

In my opinion, this study is very interesting, there are no any substantial defects, but there are several points that would need improvement or verification. I also have a few questions, because maybe I did not understand everything properly or it seems to me that some elements should be better explained.

1.       In section 2.1 you write: “…the 30‐m version is available, SRTM‐90 data will be more robust on elevation value due to lower resolution, which is beneficial for slope calculation.” - Why is that? Can you explain it more clearly? Have you tested it for your model? If so, what were the results?

2.       Line 148 – I think you should explain here at least in one sentence, what SSD and SSM are. I mentioned earlier, that when you following the model's operation scheme, it is a bit difficult to understand the procedure when some components will be discussed later (for example, in the next paragraph, you great explain what AGA is).

3.       Lines 153-154 - I miss here explaining what the DSP is? It appears in the diagram (Fig. 2).

4.       Line 217 - Can you explain this tiny difference of elevation more clearly?

5.       Figure 5 – In the figure “S” is not signed.

6.       Section 3 – Model Validation - I think that for better illustrated considered situations, it would be good to include figures showing both lakes, eg orthophotomaps (with drawn contour lines) or topographic maps with marked lake boundaries and islands. For the readers who do not know studied areas it would be easier to compare terrain situation with model results.

Description of the experiment for Ontario Lake is very well shown and the results sufficient explained. I like this part of the text, everything here is clear.

7.       Line 320 - When you write, that the modeling results well illustrate the islands on the lake, it would be good to see it on the orthophotomap or topographic map of the lake area.

8.       Line 335 – What is HydroLake [1]? Should be Hydroweb?

This study provides a completely prepared methodology for estimating lake bathymetry using a new DLBM model. The authors discussed in detail the structure and operation of the model, conducted interesting experiments and described results very well. The advantages and weaknesses of the method were indicated. I think that the model can be further developed based on better input data.

I have no objections to the content of this work. I suggest changes in the structure of the manuscript (middle part – Data and model development). After minor corrections, I recommend the article for publishing.

Author Response

Thanks for your professional comments, please refer to the attatched file.

Reviewer 2 Report

Overall comments:

This is an interesting manuscript that addresses an important topic. The paper is well written and organized for the most part. The methodology makes sense and is well laid out in section 2. However, the results section is lacking any form of quantitative measures of error/uncertainty. Without listing these results, this cannot really be referred to as “model validation”. Additionally, I am concerned with the portion of the results regarding Lake Namco. The model results seem to be an exact replica of the observed hydroweb data, which makes me question the results. Overall, this section needs to be more descriptive and do a better job of explaining and justifying the results.

Additional minor edits and additions that would strengthen this manuscript are listed below. The introduction/discussion sections are fairly short, but this seems appropriate as the primary focus of the manuscript is the new method. However, some additional references on what has been done would help strengthen the introduction.

Specific comments:

Line 55: An important method that was left out of the list is using satellite imagery/altimetry to make lake volume estimates. This is a pretty simple approach and has been done in a number of previous studies (e.g., Kropacek et al., 2012; Song et al., 2014; Keys & Scott, 2018, etc.). This method isn’t quite as accurate but is an important method to list here because it also uses entirely remotely sensed data. This is also the method that is used by hydroweb, which is utilized for validation in section 3 of this manuscript.

Line 102: Does this method only work for large lakes? You say that other DEM data can be used…did you test other DEMs of varying resolution? In other words, will this methodology work for a small lake using a fine-scale DEM (e.g., a 1-m lidar derived DEM)?

Line 109: I think these are fair assumptions. Is there any literature that supports these?

Line 138/141: How is water surface identified without a real lake bathymetric map? Is it manually delineated using the DEM?

Line 146: What is the filter function used to delete incorrect pixels?

Line 167: Need to clarify that the four directions are from an aerial/XY perspective. Up and down makes it sound like the elevation or Z-value is being examined.

Line 178: What is meant by “until any CCP occurs”?

Figure 4 caption: What is ‘vertical view’? Do you mean plan view?

Line 218/231: What causes this difference? “…due to the raster format of the data”? This needs to be further explained.

Figure 5/Line 241: The y should be a z as it corresponds to the vertical direction (elevation). X and y generally correspond to longitude and latitude, respectively.

Line 225: Need to rephrase. Something along these lines: “…, both assumed to be equal to hi due to small k0i values”

Line 227: S is not shown on the figure. It looks like it may have gotten cut off or cropped out.

Line 229: “Widths” should be “depths”

Secion 3: This section is missing quantitative validation information. Figure 7 shows curves of the modeled results compared with measured data but there is no quantitative measure of validation. Similarly, Figure 9 shows area and volume curves without any supporting statistics. Some simple descriptive statistics that measure goodness-of-fit/uncertainty such as R2, RMSE, NSE, MAE, 1:1 plots, etc. would greatly improve this section. The data is there, it just needs to be further analyzed.

Line 345: 0.03%?

Figures 9 and 10: I’m skeptical about the goodness-of-fit with the hydroweb time series. Lines 332-340 discuss the discrepancy between the two methods of estimating lake area, but somehow there is a perfect fit between the two datasets. Because the levels of accuracy between Landsat and SRTM are very different and should lead to differing results, I find it hard to believe that an 18 year time series of data would match up perfectly with a regression coefficient of 0.9997.

Second, there is a lot of uncertainty with hydroweb estimates and there is likely some uncertainty from your model, so how is it possible that there is a regression coefficient of 1.000 between these two independent datasets? It looks like the wrong data was plotted and the hydroweb data was just shifted down. I would encourage you to rerun this analysis and reexamine these results. If these results are correct, then there needs to be some justification for such a good model fit because this raises a lot more questions than it answers.

Figure 9c: Y-axis should be relative volume (i.e. ΔV), right?

Figure 10a: Is this the original modeled data? Shouldn’t there be a plot of the model fitted data vs real area series that is closer to the y=x line?

Figure 10: Also, need to remove ‘x’ and ‘y’ from the axis titles.

Author Response

Thanks for your professional comments, please refer to the attached file.

Round 2

Reviewer 2 Report

The authors have made substantial changes and the manuscript is looking much better now. Specifically, the addition of error measurements and explanations greatly strengthened the results section. Overall, I have no further comments/suggestions.